# Ligand-Based Design of Selective Peptidomimetic uPA and TMPRSS2 Inhibitors with Arg Bioisosteres

**DOI:** 10.3390/ijms25031375

**Published:** 2024-01-23

**Authors:** Patrick Müller, Collin Zimmer, Ariane Frey, Gideon Holzmann, Annabelle Carolin Weldert, Tanja Schirmeister

**Affiliations:** Institute of Pharmaceutical and Biomedical Sciences, Johannes Gutenberg University Mainz, Staudinger Weg 5, D-55128 Mainz, Germany; muelpat@uni-mainz.de (P.M.); cozimmer@uni-mainz.de (C.Z.); arfrey@uni-mainz.de (A.F.); gholzman@students.uni-mainz.de (G.H.); anwelder@uni-mainz.de (A.C.W.)

**Keywords:** trypsin-like serine proteases, covalent reversible inhibitors, enzyme inhibition study, protease inhibitors, peptidomimetic sequence, arginine bioisosteres

## Abstract

Trypsin-like serine proteases are involved in many important physiological processes like blood coagulation and remodeling of the extracellular matrix. On the other hand, they are also associated with pathological conditions. The urokinase-pwlasminogen activator (uPA), which is involved in tissue remodeling, can increase the metastatic behavior of various cancer types when overexpressed and dysregulated. Another member of this protease class that received attention during the SARS-CoV 2 pandemic is TMPRSS2. It is a transmembrane serine protease, which enables cell entry of the coronavirus by processing its spike protein. A variety of different inhibitors have been published against both proteases. However, the selectivity over other trypsin-like serine proteases remains a major challenge. In the current study, we replaced the arginine moiety at the P1 site of peptidomimetic inhibitors with different bioisosteres. Enzyme inhibition studies revealed that the phenylguanidine moiety in the P1 site led to strong affinity for TMPRSS2, whereas the cyclohexylguanidine derivate potently inhibited uPA. Both inhibitors exhibited high selectivity over other structurally similar and physiologically important proteases.

## 1. Introduction

With over 600 different proteins, proteases represent an important class of enzymes [1]. Approximately one-third of all known proteolytic enzymes are serine proteases [2]. According to the MEROPS database of peptidases, these enzymes are classified into clans by their catalytic mechanism and into families on the basis of a common ancestry [3]. The largest family of serine proteases are the trypsin-like proteases (TLPs). The catalytic triad of TLPs harbors a nucleophilic serine residue in combination with aspartate and histidine, which increase the nucleophilicity of the serine. The trypsin-like substrate specificity is characterized by the positively charged side chain of arginine or lysine in the P1 position [3,4]. Numerous important physiological processes rely on trypsin-like serine proteases. This includes hemostasis, the immune response system and extracellular matrix remodeling [5,6,7,8]. Dysregulation of these enzymes can lead to severe pathological incidents, which range from cardiovascular disorders to cancer progression or neurodegenerative and inflammation processes [8,9,10]. Moreover, proteases often are virulence factors in infectious diseases. As an example, tropical and subtropical countries are heavily affected by dengue virus infections, where the viral NS2B-NS3 trypsin-like serine protease is essential for the replication process of the virus [11]. Undoubtedly, this class of enzymes includes promising targets in various diseases, and the scientific community still strives to discover more drug candidates [7].

The urokinase-type plasminogen activator (uPA) is one member of the trypsin-like serine proteases. The enzyme is involved in the fibrinolytic system [12]. The binding of uPA to its specific glycolipid-anchored uPA receptor (uPAR) on cell surfaces enables the conversion of plasminogen to the serine protease plasmin [13]. This mediates extracellular proteolysis and the activation of several further proteases, like activating growth factors and metalloproteases, which catalyze the degradation and remodeling of extracellular matrix components [14,15]. Unfortunately, pathophysiological mechanisms like tumor angiogenesis, tumor progression and metastasis profit from these events, and therefore, inhibition of this protease could be beneficial for the mitigation, or even prevention, of tumor proliferation (Figure 1 left side) [14,15]. Blocking of the catalytic activity was achieved by specific antibodies, overexpression of the endogenous inhibitors PAI-1 and small-molecule inhibitors [16,17,18]. One of the most promising peptidomimetic inhibitors, mesupron^®^ (upamostat, WX-671, RHB-107, Wilex AG, Heidelberg, Germany), led to reduced metastasis and extended lifespan in clinical trials on pancreatic and breast cancer patients [19]. Hence, uPA can be considered as a promising drug target to block tumor dissemination.

Another proteolytic enzyme that belongs to the trypsin-like serine proteases is the human transmembrane protease serine subtype 2 (TMPRSS2). It has been shown to play an important role for viral host cell entry, and received increased attention during the SARS-CoV-2 pandemic due to its ability to enable cell entry and spread of the coronaviruses SARS-CoV-2, SARS-CoV and MERS-CoV [20,21,22,23,24]. The entry of these viruses is mediated by the spike protein, which is located at the viral cell surface. TMPRSS2 processes the spike protein after binding of the virus to the angiotensin-converting enzyme 2 receptor (ACE2), initiating the entry into lung cells (Figure 1 right side) [23,25]. Additionally, viral cell entry can occur via the endosomal pathway, whereby the spike protein is processed by cathepsin L [26]. Studies have demonstrated that inhibition of TMPRSS2 blocks the viral host cell entry and replication of SARS-CoV-2 in lung epithelial Calu-3 cells [27,28]. Previous work, in cooperation with Mailänder et al. showed that peptidomimetic inhibitors efficiently reduce TMPRSS2 activity, block SARS-CoV-2 spike-driven entry and prevent SARS-CoV-2 infection in CaCo-2 cells [29]. This highlights the opportunity for an alternative therapeutic strategy, besides targeting of the viral host proteases papain-like protease (PL^pro^) and the 3C-like- or “main protease” (3CL- or M^pro^) [30,31,32].

In the past decades, several uPA inhibitors have been disclosed, most of them with non-covalent reversible or covalent-irreversible inhibition mode [16,33]. On the contrary, only few covalent-reversible inhibitors are found in the literature [34]. Such inhibitors could combine the benefits from both concepts: the high-affinity properties and extended residence time by covalent modification of the catalytic serine residue and the reduced risk for unwanted side effects and toxicity by a reversible binding mechanism [35,36,37]. Furthermore, in order to minimize the risk for side effects, it is of great importance to inhibit the target protease selectively. This, however, is a major challenge due to the high structural similarity within the trypsin-like serine protease family. 

In 2021, the group around Huang et al. created a homology structure model of the TMPRSS2 serine protease domain, and revealed a high similarity between the homology model and the structure of the uPA [38]. This led to the idea to transfer the design of the synthesized uPA inhibitors to the previously published TMPRSS2 inhibitors, to receive an improved set of inhibitors in terms of off-target selectivity (Figure 1) [29].

Herein, we describe the ligand-based development of peptidomimetic inhibitors, which started with Ac-Gly-l-Thr-l-Ala-l-Arg-ketobenzothiazole (kbt) as a covalent-reversible uPA inhibitor discovered in previous work [39]. We substituted the P1-arginine moiety with a variety of bioisosteres, inspired by the serine protease inhibitor camostat, and furthermore modified the benzothiazole structure [40,41]. The cyclohexyl-and phenylguanidine moiety presented the most promising results during the enzyme inhibition studies. Therefore, we translated this structure motif to the suitable peptide sequence Ac-l-Asn-l-Pro-l-Arg-kbt from our previous work towards TMPRSS2 [29]. Within this study, we successfully enhanced the affinity and selectivity for both main-target proteases by systematic variation of different structural elements.

## 2. Results

### 2.1. Chemistry

All tested peptidomimetic inhibitors were synthesized in multistep reactions. First, the peptide sequences (P2–P4) of the inhibitors were prepared via a standard fmoc solid phase peptide synthesis (SPPS) protocol, which is described in detail in the Appendix A. The P1 derivatives with the ketobenzothiazole moiety as warhead were prepared as described in Figure 1 and Figure 2.

#### 2.1.1. Synthesis of the (Homo)arginine-Based Inhibitors

Boc-protected *N*_ω_-2,2,4,6,7-pentamethyl-dihydrobenzofuran-5-sulfonyl (pbf)-l-arginine **4**, which was used as the starting material for the arginine-based inhibitors **14a**–**f** and **15**, was modified to the Weinreb amide **5**. The ketobenzothiazole derivatives **12a**–**f** and **13** were obtained by alkylation of **5** with the respective heterocycles **10a**–**f** and **11**. The benzothiazole **10a** and the benzothiophene **10f** were commercially available, whereas the 6-fluoro-, 6-chloro-, 6-bromo-, 6-methoxybenzothiazoles **10b**–**e** and 4,5,6,7-tetrahydrothiazole **11** had to be synthesized by desamination of the commercially available 2-amino precursors **6b**–**e** and **9**. The 2-amino-4,5,6,7-tetrahydrothiazole **9** was prepared from cyclohexane **7** and thiourea **8** with iodine. The preparation of the homoarginine inhibitor **20** started with the guanylation of boc-protected l-lysine **16** with *N*,*N*’-bis-(carbobenzoxy)-1-*H*-pyrazole-1-carboxamidine, yielding compound **17**. Afterwards, **17** was converted to the ketobenzothiazole **19**, in analogy to the arginine derivatives. After boc-deprotection of the amino group, the P1 precursor derivatives were coupled with the Ac-Gly-l-Thr(O*^t^*Bu)-l-Ala-OH peptide **3** using 1-[bis(dimethylamino)methylene]-1*H*-1,2,3-triazolo [4,5-b]pyridinium 3-oxide hexafluorophosphate (HATU) as the coupling reagent. Final deprotection of the (homo)arginine and threonine side chain under acidic conditions and purification via RP-HPLC yielded the inhibitors **14a**–**f**, **15** and **20**.

#### 2.1.2. Synthesis of the Phenyl/Cyclohexylguanidine-Based Inhibitors

The preparation of the *p*-phenyl- and *p*-cyclohexylguanidine-based inhibitors both started with boc-protected *p*-nitro-l-phenylalanine **24**. The reduction of the nitro group was carried out with 5% Pd/C in methanol to yield **25**, whereas the hydrogenation of the benzene ring and the nitro group using the Adam’s catalyst under acidic conditions yielded the cyclohexane derivative **34**. The amine group of both compounds was guanylated with *N*,*N′*-bis-(carbobenzoxy)-1-*H*-pyrazole-1-carboxamidine. The bis-cbz-protected intermediates (**26**, **35**) were converted, in a similar way to the arginine-based inhibitors (Section 2.1.1), to the Weinreb amides **27** and **36** and later to the ketobenzothiazole derivatives **28** and **37.** After removal of the boc-protecting groups, the respective peptide sequences **3**, **21**–**23**, which were synthesized via a standard fmoc solid-phase synthesis (SPPS), were coupled with the *p*-phenyl- and cyclohexylguanidine precursor derivatives. After final deprotection of the side chains in TFA/DCM and purification via RP-HPLC, the inhibitors **29**–**33** and **38**–**39** were obtained. Starting with *m*-nitro-l-phenylalanine **40**, the inhibitor **45** was prepared in analogy to the 5-step synthetic process of the *p*-phenylguanidine derivatives **29**–**33**. The synthesized final compounds **14a**–**f**, **15**, **20**, **29**–**33**, **38**–**39**, **45** showed two peaks with identical *m*/*z* ratio and similar retention times in initial chromatographic analyses. This is due to the partial epimerization of the *α*-carbon in the P1 amino acid portion during the reaction of the Weinreb amide with lithium-benzothiazole solution. Since the faster eluting epimer was always isolated via RP-HPLC in very large excess, while the other diastereomer was obtained only in traces, we supposed the first one to be the l-epimer, and used it for all inhibition studies [42].

### 2.2. Enzyme Inhibition Studies

The inhibitory activity of the synthesized compounds towards the respective main- and off-target proteases was measured via fluorometric and colorimetric assays. Thus, fluorogenic AMC- or colorimetric *p*NA-based substrates with a peptide sequence suitable for the tested protease were utilized (see Appendix A). At first, the compounds were screened against five proteases (uPA, TMPRSS2, matriptase, tPA, thrombin, factor Xa) at 20 µM, and a cut-off value of 80% inhibition at this concentration was set, for the differentiation between nonactive (n.a.) and active inhibitors. Due to the reversible inhibition mechanism of the ketobenzothiazole derivates, the IC_50_ values were determined with Graphpad Prism 9, and afterwards converted to the corresponding *K*_i_ values for an adequate comparison between the inhibitory activities of the compounds toward all tested proteases. The *K*_i_ values were calculated using the Cheng–Prusoff equation [43].

#### 2.2.1. Inhibition Studies with uPA Inhibitors

At first, we investigated the selectivity profile of the starting compound **14a,** which exhibited good inhibition of uPA with a *K*_i_ value of 141 nM. **14a** was originally synthesized for the analysis of reactivity and selectivity studies of peptidomimetic covalent inhibitors [40]. The peptide sequence Ac-l-Gly-l-Thr-l-Ala-l-Arg was used, because of its literature-known selectivity for uPA vs. tPA [44]. Due to the similar and important physiological roles of uPA and tPA, a good selectivity is necessary to avoid severe side effects concerning ECM degradation and cell proliferation [5,45,46]. Furthermore, the trypsin-like serine proteases thrombin and factor Xa were chosen because of their important roles in blood coagulation, as well as matriptase as a representative of a transmembrane protease, which is involved in the remodeling of plasma membranes and other lipid matrix formations [47,48,49]. Due to their structural similarity (calculated sequence similarity is given in Table 1, Table 2 and Table 3) to uPA and their physiological roles, they resemble important off-targets. As expected, **14a** did not show inhibition of tPA, and only moderate selectivities for uPA towards thrombin (**14a** *K*_i_ = 4390 nM) and factor Xa (**14a** *K*_i_ = 3360 nM) with inhibition constants in the low micromolar range. In contrast, a lower *K*_i_ value was obtained for mapriptase (**14a** *K*_i_ = 32 nM). Exchanging the arginine side chain with a *p*-phenyl- or cyclohexylguanidine moiety enhanced the inhibitory properties. Both derivates resulted in more affine inhibitors (**29** *K*_i_ = 29 nM, **38** *K*_i_ = 39 nM), with a significant improvement in their selectivity profiles. The inhibitors **29**, **38** did not inhibit tPA, thrombin and factor Xa, and the selectivity indices for matriptase (**29**
*K*_i_ = 132 nM, **38**
*K*_i_ = 626 nM) were improved. The inhibitors **20** and **45**, which contain the homoarginine and *m*-substituted phenylguanidine moiety, did not show inhibition of all tested proteases at 20 µM, which highlights the importance of the alkyl chain length and the *p*-position of the guanidine element for proper binding into the S1 pocket. Additionally, all compounds were tested against the TMPRSS2 because of the aforementioned structure similarity to uPA [38]. The results indicated a strong affinity to the TMPRSS2 protease with *K*_i_ values in the nanomolar range of the arginine, phenyl- and cyclohexyl derivates (**14a** *K*_i_ = 5 nM, **29** *K*_i_ = 10 nM, **38** *K*_i_ = 73 nM). Based on these results, a SAR study with the phenyl- and cyclohexylguanidine moiety as arginine bioisosteres for new TMPRSS2 inhibitors was performed, which is described in Section 2.2.2 [29].

Besides the arginine replacement in the P1 position, we also evaluated the influence of modifications of the benzothiazole moiety (cpds. **14b**–**f**, **15**). The introduction of the electronegative halogen atoms fluorine, chlorine and bromine in position 6 led to an approximately two-fold increase in the affinity for the chloro- and bromo-derivates (**14c** *K*_i_ = 82 nM, **14d** *K*_i_ = 60 nM), and a three-fold loss of affinity for the fluoro-derivate (**14b** *K*_i_ = 388 nM). Other modifications, like the electron-donating methoxy group in position 6 (**14e** *K*_i_ = 178 nM), the exchange of the benzene ring system with a cyclohexyl ring (**15** *K*_i_ = 435 nM) or the replacement of the benzothiazole with a benzothiophene ring, led to a decrease in or complete loss of the affinity towards the uPA. Selectivity studies were performed with the chloro- and bromo-derivates **14c**–**d**, because they were the only inhibitors with slightly better affinity than the nonsubstituted ketobenzothiazole inhibitor **14a**. They revealed similar affinity to TMPRSS2 (**14c** *K*_i_ = 9 nM, **14d** *K*_i_ = 6 nM) and matriptase (**14c** *K*_i_ = 59 nM, **14d** *K*_i_ = 38 nM), and reduced selectivity vs. thrombin (**14c** *K*_i_ = 456 nM, **14d** *K*_i_ = 450 nM) and factor Xa (**14c** *K*_i_ = 2447 nM, **14d** *K*_i_ = 2847 nM) in comparison to **14a**.

#### 2.2.2. Inhibition Studies with TMPRSS2 Inhibitors

The selection of the peptide sequence Ac-l-Asn-l-Pro-l-Arg was based on the results of previous work. The published inhibitor Ac-l-Asn-l-Pro-l-Arg-kbt showed *K*_i_ values in the single-digit nanomolar range (*K*_i_ = 2.5 nM) and a good selectivity vs. thrombin (*K*_i_ = 1046 nM) [29]. Unfortunately, only slight selectivity could be observed over factor Xa (*K*_i_ = 41.1 nM), and almost no difference in inhibition potency between TMPRSS2 and matriptase (*K*_i_ = 5.2 nM). Therefore, we tried to improve the selectivity profile by substituting the arginine side chain in the P1 position with the previously used phenyl- and cyclohexylguanidine moiety. Both derivates **30**, **39** showed an increase in selectivity for TMPRSS2 towards matriptase, with the phenylguanidine-based compound being more affine for TMPRSS2 (**30** *K*_i_ = 5 nM) than the cyclohexyl derivate (**39** *K*_i_ = 44 nM), but also showing a better inhibition of the matriptase for the phenylguanidine derivate (**30** *K*_i_ = 60 nM, **39** *K*_i_ = 1198 nM). **30**–**33** and **39** did not inhibit tPA, thrombin and factor Xa. In addition, **30** and **39** showed a moderate selectivity for TMPRSS2 over uPA (**30** *K*_i_ = 479 nM, **39** *K*_i_ = 936 nM). Based on these results, and due to the overall good affinity and selectivity parameters, we decided to maintain the phenylguanidine moiety in the P1 position and implement P2 modifications with phenyl- and cyclohexylalanine (Phe, Cha) instead of proline. The latter is based on results obtained with hepsin inhibitors from the group of Kwon et al. [40]. The inhibitor **31** with the P2 phenylalanine residue showed inhibition of TMPRSS2 in the subnanomolar range (**31** *K*_i_ = 0.4 nM) and a significant increase in selectivity over matriptase (**31** *K*_i_ = 252 nM) and uPA (**31** *K*_i_ = 3574 nM). The inhibitor **32** with the cyclohexylalanine residue in P2 position also showed very good selectivity over matriptase (**32** *K*_i_ = 3333 nM) and uPA (**32** *K*_i_ = 2688 nM), but less affinity to TMPRSS2 (**32** *K*_i_ = 34 nM). In an attempt to improve the drug-like properties of the designed inhibitors, we synthesized the shortened compound **33**. This led to a slightly less active TMPRSS2 inhibitor, but still in the low nanomolar range (**33** *K*_i_ = 5 nM). The selectivity profile for TMPRSS2 inhibition over matriptase (**33** *K*_i_ = 1443 nM) and uPA (**33** *K*_i_ = 5264 nM) is still very promising. 

### 2.3. Parallel Artificial Membrane Permeation Assay (PAMPA)

Following the envisioned applications of the presented inhibitors as drug leads for the treatment of cancer or viral infections, cell permeability is an important factor in the characterization process. Since both main targets (uPA and TMPRSS2) are membrane-located, extracellular structures, inhibitors seemingly do not require cell permeation to address their target. However, in the organismic contexts of oral bioavailability (facilitated application) and biodistribution (reaching target tissue), adequate permeation is an important quality. To assess this characteristic, PAMPA was used as a suitable model for passive permeation.

Generally, the inhibitor scaffold combines some favorable features: The Arg-like P1 amino acid (in combination with other hydrophilic amino acids like Asp and Thr) ensures high aqueous solubility, even in the presence of the rather hydrophobic benzo-heteroarenes. The latter motif conveys reliable detectability by spectroscopy-based methods (λ_max_ = 305–350 nm, depending on substitution pattern). The inhibitors also were found to be sufficiently stable in the utilized aqueous system (50 mM TRIS, pH = 7.4) over the course of the assay (7 h at room temperature) and under elevated temperature conditions (17 h at 37 °C). Computed physicochemical properties, absorption spectra and stability studies are depicted in Appendix A.

However, all presented compounds were found to have very low permeabilities (P_e_ < 1 × 10^−6^ cm/s) without any indication of improvement between the structural modifications (as exemplified for **38** in Figure 2). This result is not surprising. The pK_a_ (of the protonated guanidine function) of all compounds is calculated to be ≥10 (Marvin JS 23.11.0), meaning that in assay (or physiologic) conditions, all compounds are expected to be fully (≥99.75%) protonated, and therefore remarkably hydrophilic. Most of the presented compounds have negative logD_7.4_ values, with **33** being the exception (logD_7.4_ = 1.2; compare Appendix A). This level of lipophilicity, however, was still not enough to exert measurable permeability. For approved drugs with similar structural characteristics (e.g., camostat, melagatran, xylometazoline, metformin), only very limited permeabilities are described as well [50,51,52,53]. All this indicates the pronounced hindering effect of the guanidine group for passive permeation.

The discussed properties of the presented compounds can be paralleled to BCS class III compounds, namely their high aqueous solubility and low permeability. For these types of drugs, one major option to improve permeability is to remove charge from the molecule. In amidine-containing drugs, where charge is almost pH-independent due to their immense basicity, this was addressed by conversion to the amidoxime (ximelagatran or mesupron^®^) or carbamate prodrugs (dabigatran) with lower basicities [19,50,54]. For the guanidine moiety, the conversion to *N*-hydroxyguanidine is possible [55]. In a technological approach to improved absorption, possible options for oral application are the formulation with permeation enhancing agents, or lipophilic counter ions [56]. For intravenous applications, nanoparticular formulations can be applied (e.g., for doxorubicin or for protease inhibitors) [57,58]. Of course, combinations of both chemical and technological approaches should be employed for optimization.

## 3. Discussion

Trypsin-like serine proteases present attractive drug targets for treatment against many diseases, which can be of malignant cellular or viral origin [9,11]. Over the past decades many potent inhibitors were designed with remarkable affinity for the target protease. But most of them lack selectivity because of the highly structural similarity between the proteases. Within our study, we describe a systematic ligand-based approach to enhance affinities and selectivities. Starting from the previous published covalent reversible ketobenzothiazole inhibitor Ac-Gly-l-Thr-l-Ala-l-Arg-kbt **14a**, we modified the P1 arginine side chain with different bioisosteres [39]. The results indicate that the cyclohexylguanidine moiety fits best for uPA inhibition. The inhibitor **38** showed remarkable inhibition with a *K*_i_ value of 39 nM and a very good selectivity profile towards the other trypsin-like serine proteases. The modification of the benzothiazole moiety did not improve either the inhibitory properties nor the selectivity profiles, rendering the original kbt warhead the most promising.

The transfer of the P1-arginine replacement with the promising phenyl- and cyclohexylguanidine moieties to the previously published TMPRSS2 inhibitor Ac-l-Asn-l-Pro-l-Arg-kbt was a success, leading to a subnanomolar TMPRSS2 inhibitor **31** (*K*_i_ = 0.4 nM), with significantly increased selectivity over other trypsin-like serine proteases [29]. Furthermore, the shortened peptide sequence of the TMPRSS2 inhibitor **31** led to the more drug-like candidate **33,** with still very good inhibitory and selectivity properties. In terms of permeability, the inhibitor scaffold (and especially the shortened compound **33**) leaves the opportunity for improvement in a focused structure-permeability relationship study. 

## 4. Materials and Methods

The materials as well as the methods used for this study are described in the Appendix A. The authors have cited additional references within the Appendix A [29,39,40,43,59,60,61,62,63,64,65,66,67,68,69,70,71,72]. Appendix A of the protein similarity calculation (Appendix A), fluorometric inhibition assays (Appendix A), absorption spectra (Appendix A), stability studies (Appendix A), NMR-spectra and HPLC-chromatograms (Appendix A) and Appendix A of the computation of physicochemical parameters can be accessed in the Appendix A.

## Data Availability

Data is contained within the article and Appendix A.

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
