# Peer review of "Ligand-Based Design of Selective Peptidomimetic uPA and TMPRSS2 Inhibitors with Arg Bioisosteres"

_ijms, 2024, doi:10.3390/ijms25031375_

Round 1

Reviewer 1 Report

Comments and Suggestions for Authors

The work of Muller and co-workers is relevant and interest for health research audience/readers. The work was well conducted and presented, so I support its publication after minor adjustments.

I think readers will understand better some affirmations after graphical comparison or visualization. Thus, I suggest the authors to add an Figure with the 3D superposition of uPA and TMPRSS2 structures. Likewise, an LigPlot scheme, showing clearly the active site and the relevant aminoacids would be helpful, perhaps using the original peptidomimetic as reference ligand. 
In addition, the structural and/or sequence similarity with the other proteases (tPA, matriptase, thrombin) could be demonstrated either by estimating the % of sequence similarity (using clustalOmega or other) or by superimposing the structures of the different enzymes. This will make the discussion in section 2.2.1 even more complete.
I'd like to see some discussion about the fact that 38 is more promising. While the group is bulkier, the simple connections allow for some flexibility, which may be why it works better.
Finally, the authors could suggest alternatives to overcome the permeability issue. What kind of chemical alterations could be made to the peptide to achieve an amphipilic balance? Or could the peptide be transported by a vehicle?

Author Response

Dear Reviewer 1:

Thank you for your revision.  

We implemented a graphical superimposition of the uPA and TMPRSS2 protein structures in Figure 1 for better visualization of the mentioned structure similarity (page 3). To underline this for all investigated proteases, we added the calculated binding site sequence similarities in table 1, 2 and 3, and a visual representation of this to the SI.

While we agree with your reasoning on a possible difference due to the higher conformational flexibility of 38, we chose not to claim any reasoning, because the selectivities of both inhibitors 38 and 29 are in the same range. In our opinion, a true difference from structural implications has to be proven by a crystallographic structure analysis that is not in the scope of this paper. However, we decided to choose 38 as the lead compound after our systematic study to present a conclusive lead structure for further drug development.

To present more diverse suggestions for permeability improvements, an alternative prodrug modification and possible technological solutions for drug formulation were added (page 11, lines 322333).

Thank you for your time and effort. 

Best,

Tanja Schirmeister

Reviewer 2 Report

Comments and Suggestions for Authors

This manuscript describes a ligand-based appraoch to the development of reversable, noncovalent peptidomimetic inhibitors of the Trypsin-like serine proteases uPA and TMPRSS2. Starting with Ac-Gly-L-Thr-L-Ala-L-Arg-ketobenzothiazole (kbt) as a covalent-reversible uPA inhibitor, the P1-arginine was changed to other bioisosteres, inspired by the serine protease inhibitor camostat. Some further modifications were then made to the peptides, resulting in some noncovalent inhibitors with good potency. The affinity and selectivity for both proteases was enhanced by systematic variation of different structural features of the inhibitors, and potent inhibitors were found for both uPA and TMPRSS2. The inhibitors seem to have issues crossing cell membranes however, likely due to the unprotected guanidine, which when protonated results in a highly hydrophilic compound. This may be addressed by conversion to the amidoxime, which is often used in other amidine-containing drugs, but the authors leave this for future work. Overall this is a good example of ligand based inhibitor design, and should interest many medicinal chemists and biochemists. I did find a few minor issues which should be looked at before publication, including the following:

• Line 30 "the MEROPS database..." this database is unfamiliar to me, so it will probably be unfamiliar to many readers, therefore it's probably better to say "the MEROPS database of pepsidases..."

• Line 32, " The catalytic triad harbors..." should probably read "The catalytic triad of TLPS harbors...", otherwise it's unclear what catalytic triad is being referred to.

• The text in figure 1 is small, perhaps the figure could be reorganized so that the panels are in a vertical arrangement above one another so the pictures can be expanded and the text made larger?

• Line 285 " under elevated conditions" should read " under elevated temperature conditions"

• In the Discussion section, the compound numbers should be in bold

Author Response

Dear Reviewer 2: 

Thank you for your revision.  

All mentioned suggestions for improvement were considered and adopted. We increased the letter size by one for better readability in figure 1 but would like to keep the original design.

Thank you for your time and effort. 

Best,

Tanja Schirmeister
